# Longitudinal Speckle Tracking Strain Abnormalities in Chagas Disease: A Systematic Review and Meta-Analysis

**DOI:** 10.3390/jcm11030769

**Published:** 2022-01-31

**Authors:** Sergio Alejandro Gómez-Ochoa, Lyda Z. Rojas, Juliana Alexandra Hernández-Vargas, Jorge Largo, Taulant Muka, Luis E. Echeverría

**Affiliations:** 1Institute of Social and Preventive Medicine (ISPM), University of Bern, 3012 Bern, Switzerland; taulant.muka@ispm.unibe.ch; 2Research Group and Development of Nursing Knowledge (GIDCEN-FCV), Research Center, Fundación Cardiovascular de Colombia, Floridablanca 680004, Colombia; lydarojas@fcv.org; 3Cuenta de Alto Costo, Fondo Colombiano de Enfermedades de Alto Costo, Bogotá 111711, Colombia; julianahernandezv@gmail.com; 4Internal Medicine Department, Universidad Militar Nueva Granada, Bogotá 110010, Colombia; jomalagil@gmail.com; 5Heart Failure and Heart Transplant Clinic, Fundación Cardiovascular de Colombia, Floridablanca 680004, Colombia; luisecheverria@fcv.org

**Keywords:** Chagas disease, echocardiography, speckle tracking strain

## Abstract

Background: Chronic Chagas cardiomyopathy (CCM) is ranked among heart failure etiologies with the highest mortality rates. CCM is characterized by alterations in left ventricular function with a typical and unique pattern of myocardial involvement. Left ventricle longitudinal speckle tracking strain is emerging as an important additive method for evaluating left ventricular function and risk of future cardiovascular events. This systematic review aimed to characterize the left ventricle (LV) longitudinal strain by speckle tracking patterns in the different stages of Chagas disease, compared to healthy controls. Methods: Searches in Medline, EMBASE, and LILACS databases (from inception to 20 May 2021) were performed. Articles written in any language that assessed patients with Chagas disease and reported any measures derived from the left ventricular strain by speckle tracking were included. Two reviewers independently selected the studies, extracted the data, and assessed the quality of evidence. Standardized mean differences (SMD) were pooled using random-effects meta-analyses. Results: Of 1044 references, ten studies, including a total of 1222 participants (CCM: 477; indeterminate form: 444; healthy controls: 301), fulfilled the selection criteria and were included in the final analysis. Patients with CCM had a significantly higher mean global longitudinal strain (GLS) value than indeterminate form (IF) patients (SMD 1.253; 95% CI 0.53, 1.98. I^2^ = 94%), while no significant difference was observed between IF patients and healthy controls (SMD 0.197; 95% CI −0.19, 0.59. I^2^ = 80%). Segmental strain analyses revealed that patients with the IF form of CD had significantly worse strain values in the basal-inferoseptal (SMD 0.49; 95% CI 0.24, 0.74. I^2^: 24%), and mid-inferoseptal (SMD 0.28; 95% CI 0.05, 0.50. I^2^: 10%) segments compared to healthy controls. Conclusions: Our results suggest different levels of functional derangements in myocardial function across different stages of Chagas disease. Further research is needed to assess the prognostic role of LV longitudinal strain and other measures derived from speckle tracking in CD patients regarding progression to cardiomyopathy and clinical outcomes prediction.

## 1. Introduction

Chagas disease (CD) is an infectious disease caused by the protozoan parasite *Trypanosoma cruzi* (*T. cruzi*). Nowadays, it is considered the parasitic disease with the highest attributable morbidity and mortality burden worldwide [1]. CD currently affects approximately six to seven million people in endemic countries in Central and South America, with an increasing incidence of the disease in North America and Europe [2,3,4,5]. The acute phase of this disease often goes underrecognized, as it is rarely severe; afterward, the chronic stage of the disease develops, in which the parasite persists in specific host tissues [6,7,8]. This period is called the indeterminate form (IF) of CD, which is characterized by the latency of the disease, and approximately 80% of the patients remain in this stage without exhibiting signs or symptoms of organ involvement [9]. From the IF, around 10% to 20% of patients will develop the cardiac form of the disease, named chronic Chagas cardiomyopathy (CCM) [10]. CCM represents the most common form of chronic involvement, characterized by extensive arrhythmogenic and thrombogenic status, myocardial fibrosis, segmental wall motion abnormalities, and ultimately a dilated cardiomyopathy with rapidly progressive heart failure, all of which confer high morbidity and mortality [7,11].

Echocardiography represents one of the pillars of multimodal evaluation of the CCM patient nowadays, providing critical information regarding the structural and functional status of the myocardium, with relevant therapeutic and prognostic implications [12]. Nevertheless, conventional measures such as the ejection fraction do not sufficiently reflect the intricate pathophysiological processes occurring in CD and CCM, especially in the early stages of myocardial involvement [13,14]. In this context, new echocardiographic techniques such as two-dimensional speckle-tracking echocardiography have recently opened the possibility of performing more accurate assessments of the systolic function by evaluating the deformation of myocardial fibers [15]. Furthermore, growing evidence suggests that global longitudinal strain (GLS) could be a more sensitive marker of systolic dysfunction of the left ventricle and even provide a more accurate cardiovascular prognosis than ejection fraction in several contexts [16,17]. The assessment of the patterns of left-ventricle involvement by segments using speckle tracking strain could, in addition, be critical for identifying patients with early myocardial involvement and those at risk of developing CCM [15]. Furthermore, this technique has the potential of predicting the development of new-onset arrhythmias, along with worsening heart failure [18,19,20]. Although several studies evaluating the role of longitudinal strain in CD have been published in the literature, their divergent results and the heterogeneity of the included populations preclude the possibility of firm conclusions in the absence of a systematic synthesis of the evidence [21,22].

The main objective of this systematic review and meta-analysis was to characterize the longitudinal strain by speckle tracking patterns in the different stages of Chagas disease and evaluate the differences between CD patients and healthy controls.

## 2. Methods

This systematic review and meta-analysis was conducted following a recently published guideline on how to perform a systematic review and reported following the PRISMA guidelines (Appendix A) [23]. The protocol of this study is registered in the PROSPERO database with the record code CRD42021225139.

### 2.1. Data Source and Strategy

MEDLINE (National Library of Medicine, US), EMBASE (Elsevier, Netherlands), and LILACS (Biblioteca Virtual en Salud, Brazil) were searched to identify relevant articles from inception until 20 May 2021, without language restrictions. The following search terms were used: Chagas disease, Trypanosoma cruzi, T. cruzi, chronic Chagas cardiomyopathy, echocardiography, and speckle tracking. Finally, we limited our search to human studies. The complete search strategy is described in Supplementary Box S1.

### 2.2. Study Selection and Eligibility Criteria

All observational studies (e.g., cross-sectional, cohort, case-control studies, and case-series), except for case reports, were included. We included studies that assessed adult (18+ years old) patients with Chagas disease (either CCM of IF individuals) and reported any measure derived from the left ventricular strain by speckle tracking. The included studies also had to compare the strain data from CD patients to a control group or analyze the differences between CCM and IF patients. We excluded those articles that did not clarify if the included CD patients had established myocardial involvement, those assessing other speckle tracking strain methods (circumferential, radial, etc.), and those performed in a pediatric population. Two independent reviewers screened the titles and abstracts according to the selection criteria.

### 2.3. Data Extraction

A data extraction form to collect relevant information from the included studies was created in the Microsoft Excel program (Microsoft Corporation, US, 2018). Two reviewers independently extracted the following data from each study: first author’s name, study location, study design, sample size, sex distribution, mean age, essential echocardiographic variables (LVEF, left-atrial size, LV end-diastolic diameter, LV end-systolic diameter, and E/e’ ratio), global and segmental longitudinal strain data, and intra/inter-observer reliability.

### 2.4. Risk of Bias Assessment and Quality Assessment

Two authors assessed the quality of included studies independently using Newcastle–Ottawa Scale (NOS) critical appraisal tools for cross-sectional, cohort studies, and case-control, as applicable [24]. The scale was developed for non-randomized and observational studies and assessed quality in three broad categories: selection of study groups/participants, comparability of the study groups/participants, and exposure/outcome of interest assessment. Quality was assessed on a 10-point scale and classified as good quality (9–10 points), fair quality (6–8), and poor quality (< 6).

### 2.5. Statistical Analysis

A narrative synthesis was performed for the included studies. For the meta-analysis, we pooled means for analyzing the trends of the continuous variables in each group. Moreover, we used standardized mean differences (SMD) and 95% confidence intervals (CIs) for assessing differences in the measures between groups. We used SMDs instead of mean differences due to the current limitations in interpreting strain by speckle tracking values, as there is still debate regarding the clinical relevance of variations in this measure. Physiological GLS values may be influenced by variables such as systolic blood pressure, body surface area, and sex; therefore, normal values are still debated. For the included cohort studies, only the baseline estimates were used. The inverse variance weighted method was used to combine summary measures using random-effects models to minimize the effect of between-study heterogeneity. Heterogeneity was assessed using the I^2^ index [low (I^2^ < 25), moderate (I^2^ > 25% and <75%), or high (I^2^ > 75%)]. Publication bias was appraised using funnel plots and Egger’s test for assessing asymmetry. For studies reporting only median and measures of dispersion (interquartile range, range, and maximum-minimum values), we converted these values into mean and standard deviation [25]. All tests were 2-tailed; *p*-value < 0.05 was considered statistically significant. Stata release 16 (Statacorp, Texas, US, 2020) was used for all analyses.

### 2.6. Sensitivity Analyses

The mean age, proportion of males, mean left ventricular ejection fraction, and other conventional echocardiographic measures were pre-specified as characteristics for assessing heterogeneity in the global longitudinal strain measure and were evaluated using univariate random-effects meta-regression. Finally, a sensitivity analysis was performed for the global and segmentary strain data using mean differences instead of SMDs.

## 3. Results

The initial search yielded 1044 studies, from which ten studies met the inclusion criteria (Figure 1) [15,21,22,26,27,28,29,30,31,32]. The studies were performed in Brazil (n = 7), Argentina (n = 1), Colombia (n = 1), and Spain (n = 1) from 2008 until 2019. A total of 1222 participants were evaluated in the included studies. From the total patients included in the meta-analysis, 477 (36.6%) had a serological confirmed diagnosis of CD, and evidence of myocardial involvement by the disease (CCM Group), 444 (36.3%) were classified as having the IF while the rest (n = 301; 27.1%) served as healthy controls. Most of the studies had a cross-sectional design (n = 7; 70%) (Table 1), while the rest had a prospective design. Finally, all strain measurements were performed with a General Electric (GE) Vivid ultrasound system, being the GE Vivid 7 Ultrasound Machine the most frequently used. All measurements were analyzed using the EchoPAC software and in two-dimensional mode, mainly following the American Society of Echocardiography recommendations. Four studies assessed intra-observer and inter-observer reliability, ranging from 0.69 to 0.99, while in two studies, a single observer performed echocardiographic measurements (Appendix A). Finally, only three studies described the prevalence of conduction disorders, while six studies excluded patients with pacemakers and the remaining four did not specify the prevalence of implantable devices in the evaluated population. Table 2 summarizes the pooled characteristics of the main echocardiographic variables provided in each study and the differences between different CD populations and healthy controls.

### 3.1. Longitudinal Speckle Tracking Strain in the Indeterminate form of Chagas Disease

Nine studies (444 individuals) evaluated patients with the indeterminate form of CD using speckle tracking echocardiography. We observed a pooled proportion of males of 39% (95% CI: 31%, 48%. I^2^: 66%) and a mean age of 50.4 years (95% CI: 45, 56. I^2^: 0%) in the populations included in the analysis. Regarding GLS, a pooled value of −19.9% (95% CI −21.2%, −18.7%. I^2^: 45) was observed. Mean GLS values were higher for populations including more male participants (Coef. 0.09; 95% CI 0.04, 0.15), while no impact of age and other conventional echocardiographic measures on mean GLS was observed (Appendix A). Finally, segmental strain analysis revealed that only the basal-anteroseptal, basal-inferoseptal, and apical-lateral segments had a median value higher than −18% (Figure 2 and Appendix A).

### 3.2. Longitudinal Speckle Tracking Strain in Chronic Chagas Cardiomyopathy

Eight studies (477 patients) evaluated the features of longitudinal strain by speckle tracking in patients with CCM. We observed a pooled proportion of males of 50.8% (95% CI: 44%, 58%. I^2^: 0%) and a mean age of 49 years (95% CI: 42%, 56%. I^2^: 43%) of participants included in this analysis. A pooled GLS value of −15.9% (95% CI −17.8%, −13.9%. I^2^: 78%) was observed in CCM patients. The meta-regression analysis showed that mean GLS value was higher in studies that included more males (Coef. 0.22; 95% CI 0.08, 0.36), had a lower mean LVEF (Coef. −0.32; 95% CI −0.47, −1.65), and higher left atrial volume index (Coef. 1.59; 95% CI 0.83, 2.36) and left ventricular end-systolic diameter (Coef. 0.37; 95% CI 0.10, 0.63) (Figure 3 and Appendix A). Finally, apical-anterior, apical-lateral, basal-inferoseptal, and basal-inferolateral segments had the worse pooled mean values of strain by speckle tracking (Figure 2 and Appendix A).

### 3.3. CCM vs. Indeterminate Patients

Seven studies, based on 717 individuals, compared features of longitudinal strain by speckle tracking between patients with CCM and the IF. Patients with CCM diagnosis had similar age distribution and male proportion compared to those in the IF group (Table 2). All of the evaluated echocardiographic parameters in the included population for the analysis were significantly worse in CCM patients than in individuals with the IF; SMD provided in Table 2. The meta-analysis showed that patients with CCM had a significantly higher mean value of the GLS (SMD 1.253. 95% CI, 0.53, 1.98. I^2^ = 94%); however, the heterogeneity of this result was high (94%). Meta-regression analyses showed that the mean LVEF and LV end-systolic diameter of the CCM patients were significantly associated with the SMD of the GLS observed between the two groups; the difference was higher when these echocardiographic parameters were worse (Appendix A).

On the other hand, the analysis of the segmentary strain by speckle tracking data revealed significant differences in eight (basal-inferolateral, basal-inferior, mid-anterior, apical-anterior, apical-lateral, apical inferior, and apical septal) of the 16 segments assessed. In contrast, other two (mid-anterolateral and mid-anteroseptal) showed marginal but non-significant differences between the two groups (Appendix A and Figure 2).

### 3.4. Indeterminate Patients vs. Healthy Controls

Eight studies, based on 649 individuals, compared features of longitudinal strain by speckle tracking between patients with the IF and healthy controls. Patients with the indeterminate disease form were significantly older than healthy controls (SMD 0.39; 95% CI: 0.06, 0.72. I^2^ = 72%). No significant differences in sex distribution and the evaluated echocardiographic variables were observed between IF and healthy controls (Table 1). Furthermore, although the pooled GLS value was not statistically different between the groups (mean difference 0.39; 95% CI. −0.5, 1.3. I^2^: 99%), the analyses of the individual segments of the LV revealed significant differences between the two groups. Specifically, patients in the IF group had a worse strain value in the basal-inferoseptal (SMD 0.49; 95% CI 0.24, 0.74. I^2^: 24%) and mid-inferoseptal (SMD 0.28; 95% CI 0.05, 0.50. I^2^: 10%) segments (Appendix A and Figure 4). Meta-regression analyses did not show a significant effect of other demographic and echocardiographic variables in these results.

### 3.5. Sensitivity Analysis

Finally, although most comparisons remained the same, we observed some variations regarding the statistically significant difference in segmentary strain values between groups when mean differences were used instead of SMDs (Appendix A). For example, no significant differences were observed between the CCM and IF groups for the mid-inferior, apical-lateral, and apical-inferior segments. On the other hand, patients in the IF had a significantly different value of the basal-anterior and apical-septal segments than healthy controls.

### 3.6. Quality of Studies and Publication Bias

Most studies included in the current systematic review were evaluated as medium risk of bias (n  =  7), while three were classified as at low risk of bias (Appendix A). Finally, we assessed for publication bias in the analyses of the GLS value per group by using Egger’s test and graphically with funnel plots. We did not observe a significant publication bias for the GLS value in patients with CCM (Egger’s *p*-value: 0.711), the IF of CD (Egger’s *p*-value: 0.197), and healthy controls (Egger’s *p*-value: 0.601), with evidence of symmetry in the funnel plots (Appendix A).

## 4. Discussion

The present study results show that the global longitudinal strain (GLS) value was significantly worse in patients with CCM than in those with the IF of the disease. On the other hand, there was no significant difference in the GLS value when comparing IF patients with healthy controls. Patients with CCM tended to have worse strain values in the apical-anterior, apical-lateral, basal-inferoseptal, and basal-inferolateral segments. In contrast, IF patients have a more severe involvement in the basal-anteroseptal, basal-inferoseptal, and apical-lateral segments. Furthermore, we observed that patients with the IF of CD exhibited significantly worse strain values in the basal-inferoseptal and mid-inferoseptal segments compared with healthy controls. To our knowledge, this is the first meta-analysis to comprehensively assess longitudinal strain by speckle tracking measures in Chagas disease patients.

Myocardial injury in CD has been subject to extensive study due to the particularity of the anatomical involvement and the disease’s progressive nature; however, the pathophysiological mechanisms behind these derangements are still not well known [7]. Currently, the autoimmune hypothesis that suggests the existence of serum antibodies against the myocardium, endocardium, and blood vessels is one of the most supported, as there is evidence revealing a lack of correlation between parasites presence and the extent of myocardial lesions [33,34,35]. Furthermore, microvascular derangements (decapillarization, intravascular platelet aggregation, interstitial edema, and basal membrane thickening) have also been reported due to *T. cruzi’s* presence or the immune response reaction [36]. These microvascular perfusion alterations are known to lead to ischemia-like symptoms and the characteristic regional wall motion impairments observed in CCM [36,37]. This unique pattern of myocardial involvement has been described in several studies using magnetic resonance imaging (MRI) and single-photon emission computed tomography methods, highlighting an extensive involvement of the inferior, inferolateral, and apical regions, as was observed in our meta-analysis results [38,39,40,41]. The reasons behind this specific distribution of myocardial replacement with fibrosis are still unclear; however, it has been suggested that the microvascular derangements characteristic of CCM might cause an ischemic process predominantly in distal areas of the coronary arteries (denominated as watershed zones), potentially explaining the higher frequency of lesions in the LV inferior and inferolateral walls and the apex [42]. Finally, other pathophysiological mechanisms, such as the derangements in the hypothalamic–pituitary–adrenal axis and specific pathways of iron metabolism, may play a critical role in CCM pathogenesis and progression [6,43,44].

The recent advances in echocardiographic imaging techniques such as speckle tracking strain have opened the possibility of performing a non-invasive assessment of concrete measures at a relatively low cost [45,46]. An example is myocardial deformation and its components (longitudinal shortening, circumferential shortening, and radial thickening), which may allow the detection of impaired LV mechanics even in patients without LV diastolic dysfunction or evident LV hypertrophy [47]. Consequently, there is a large body of evidence supporting the diagnostic and prognostic value of strain by speckle tracking in several cardiovascular diseases, being the GLS nowadays considered one of the most consistent and reproducible myocardial function parameters [48,49,50]. Among the advantages of GLS over other methods such as Doppler imaging highlights that it is less load- and angle-dependent, along with being less time-consuming [49]. In the context of CCM, our study represents the first systematic review and meta-analysis describing the available evidence regarding myocardial involvement patterns via longitudinal speckle tracking strain in patients with CCM, highlighting a worse segmental strain value in patients with IF disease compared to healthy controls. However, we acknowledge that this finding might have been influenced by the significantly younger mean age observed in the group of controls; as only two studies matched cases and controls by age, and only one by sex. Therefore, it is not clear whether the differences we found are confounded by these factors, and future studies on this topic should match cases and controls for relevant variables such as age and sex.

Nevertheless, recent studies have revealed that myocardial involvement in CD is not restricted to patients with CCM, as patients in the IF of the disease have also exhibited pathological findings such as myocyte degeneration and ongoing fibrosis [51,52]. The IF of CD is defined as the presence of a positive anti-*T. cruzi* serology (based on two different assays) along with the absence of symptoms or physical signs of disease, a normal electrocardiogram, and normal chest and gastrointestinal radiographic studies [1,9,53]. However, these criteria were defined almost 40 years ago, relying on relatively simple measures and heart and myocardial function assessments. More recently, the concept of “indeterminate form” has been widely debated, as several studies have reported anatomical and functional myocardial derangements in patients with this form of CD [54]. The reported functional alterations include anomalies in left ventricle contractility, atrioventricular conduction disorders, alterations in the systolic function of the right ventricle, and even LV segmental wall motion abnormalities [9,22,55].

Therefore, a new proposal for revising the “indeterminate form” concept was published in 2002, suggesting a normal echocardiogram as an additional criterion for classifying a patient in the indeterminate stage of the disease [54]. In our systematic review and meta-analysis, we observed that despite having similar basic echocardiographic parameters and a comparable GLS value, patients with the IF had a significantly worse longitudinal strain in two segments of the LV compared to healthy controls. These results may suggest the relevance of including the longitudinal strain by speckle tracking and other myocardial deformation measures in the criteria for classifying patients with CD into the different forms of the disease. However, further research is needed to assess the impact of this suggestion.

The findings of a significantly worse myocardial function have not been exclusively observed in longitudinal strain measures, as studies assessing radial and circumferential strain data have reported similar results regarding the presence of myocardial deformation derangements in patients with the IF of CD. In the study of García-Álvarez et al., patients with the IF had a significantly lower global and mid inferior radial strain values than controls (*p* = 0.046) [26]. A similar result was observed in the study of Barbosa et al., in which the radial strain value of the anteroseptal, basal anterior, basal inferolateral, basal inferior, and inferoseptal walls was significantly lower in patients with the IF of the disease compared to healthy controls [27]. However, these results have also been inconsistent, as other studies have reported similar global and segmental radial strain values between patients with the IF of CD and healthy controls [22,29]. Similar divergent data regarding circumferential strain and other myocardial deformation measures highlight the need for larger cohort studies for evaluating the role of strain by speckle tracking in the multimodal assessment of the patient with CD. Furthermore, there is a relevant need for assessing the value of GLS and segmentary strain in predicting clinically relevant outcomes beyond other echocardiographic parameters. This potential added value of strain-derived measures will facilitate an early identification of patients with a higher risk of progressing to CCM and developing adverse cardiovascular outcomes, opening the possibility of initiating early preventive measures. Nevertheless, the low availability of this technique in secluded rural areas of endemic countries poses a significant limitation for its widespread use as a standard tool for the multimodal assessment of the CD and CCM patient.

### Study Limitations

Our meta-analysis has several limitations, including the heterogeneity observed among the different study groups, especially those with the IF of CD. This finding could relate to the non-specific criteria for defining the IF of the disease, which causes a high variability of the strain values due to the different degrees of myocardial involvement in this group. This heterogeneity significantly limited our analyses; nonetheless, the meta-analyses that showed a statistically different value between the IF patients and controls had a low heterogeneity (I^2^ < 25%), highlighting the potential relevance of this finding despite the study limitations. Furthermore, despite most studies being of high quality, the data provided are mainly of cross-sectional nature. In addition, no inter- and intra-observer variability adjustments across the included studies were performed, limiting the possibility of including this relevant measure in the meta-analyses. Moreover, confounding bias cannot be ruled out in the present study, as relevant confounding factors were not considered in the unadjusted means provided. Finally, despite most studies excluded patients with pacemakers, we were not able to evaluate the role of conduction disorders in the GLS due to the small number of studies reporting the prevalence of these conditions. This represents a relevant limitation considering the high incidence of conduction disorders in CCM patients and the significant impact of these in strain by speckle tracking values.

## 5. Conclusions

Chagas disease is characterized by a unique pattern of progressive myocardial involvement assessed by longitudinal strain by speckle tracking. However, functional derangements in myocardial deformation are not exclusive of patients with chronic Chagas cardiomyopathy. Patients with the indeterminate form of the disease showed significantly worse strain values in the basal-inferoseptal and mid-inferoseptal segments than healthy controls. Further research is needed to assess the prognostic role of LV longitudinal strain and other measures derived from speckle tracking in CD patients regarding progression to cardiomyopathy and clinical outcomes prediction.

## Figures and Tables

**Figure 1 jcm-11-00769-f001:**
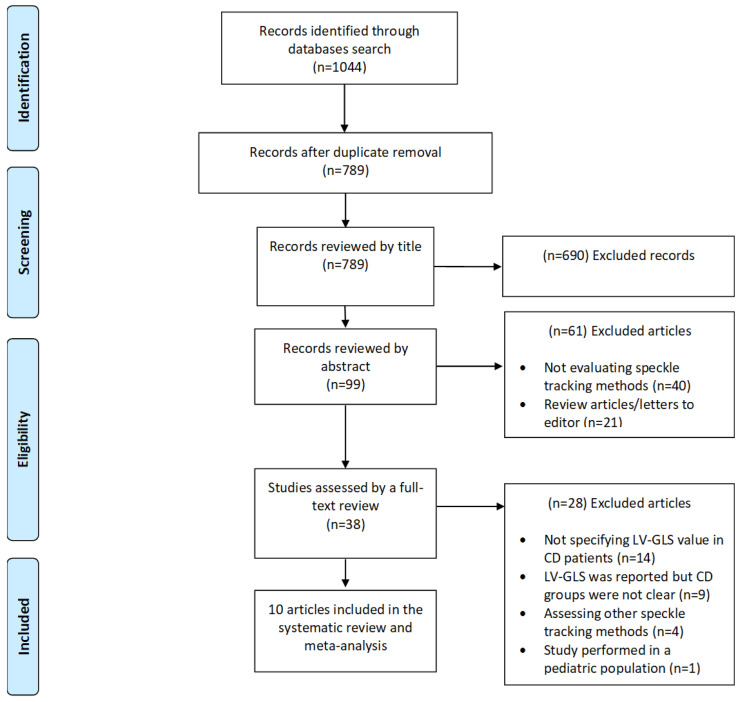
Flowchart summarizing study selection process.

**Figure 2 jcm-11-00769-f002:**
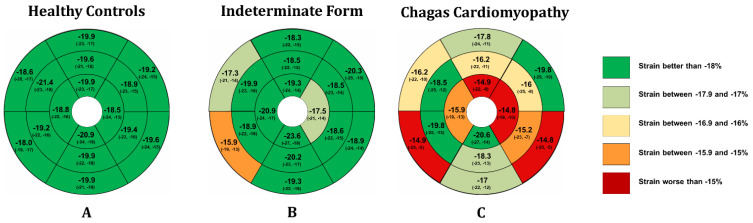
Longitudinal strain bull’s eye plot patterns in each of the assessed groups.

**Figure 3 jcm-11-00769-f003:**
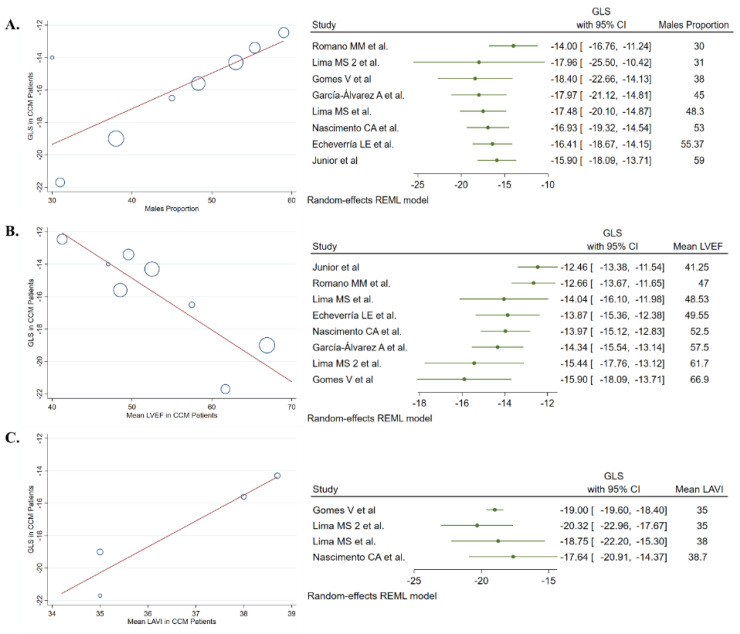
Meta-regression analyses revealed a significant association between (**A**) males proportion, (**B**) left ventricle ejection fraction, and (**C**) left indexed atrial volume with the global longitudinal strain value in patients with chronic Chagas cardiomyopathy.

**Figure 4 jcm-11-00769-f004:**
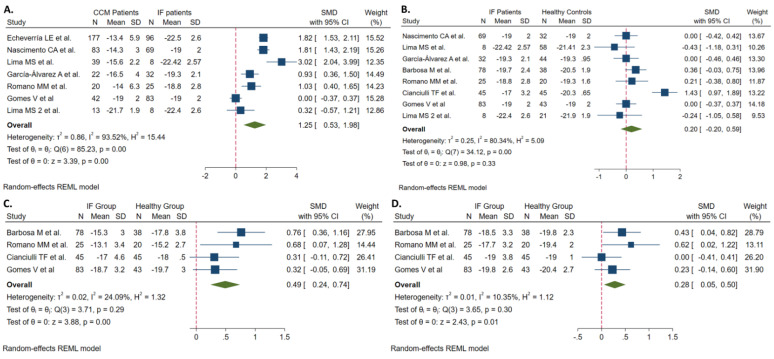
Forest plots evaluating the differences in (**A**) global longitudinal strain between CCM and IF patients, (**B**) global longitudinal strain between IF patients and controls, (**C**) basal-inferoseptal strain, and (**D**) mid-inferoseptal strain between IF patients and controls.

**Table 1 jcm-11-00769-t001:** Baseline demographic characteristics of patients assessed in the included studies.

Studies	Publication Year	Country	Total Patients	CCM Patients	Mean Age of CCM Patients	CCM Men (%)	IF Patients	Mean Age of IF Patients	IF Men (%)	Healthy Controls	Mean Age of Healthy Controls	Controls Men (%)
García-Álvarez A et al.	2011	Spain	98	22	42.7 ± 9.6	45	32	36.8 ± 10.8	34	44	34 ± 8.5	41
Nascimento CA et al.	2013	Brazil	184	83	48 ± 8.7	53	69	45 ± 9	48	32	44 ± 7	44
Barbosa M et al.	2013	Brazil	116	-	-	-	78	44.7 ± 8.6	46	38	44 ± 9.2	58
Gomes V et al.	2016	Brazil	168	42	47 ± 8	38	83	44 ± 9	46	43	44 ± 7	49
Lima MS 2 et al.	2016	Brazil	42	13	53 ± 11.6	31	8	57 ± 7.5	13	21	50 ± 6.4	33
Lima MS et al.	2016	Brazil	105	39	55 ± 11	48.3	8	56 ± 7	12	58	37 ± 12	43
Santos Junior et al.	2019	Brazil	81	81	57 ± 11	59	-	-	-	-	-	-
Echeverría LE et al.	2020	Colombia	273	177	59 ± 11	55.37	96	49 ± 11	25.19	-	-	-
Romano MM et al.	2020	Brazil	65	20	53 ± 12	30	25	55 ± 12	60	20	48 ± 10	55
Cianciulli TF et al.	2020	Argentina	90	-	-	-	45	59 ± 9.9	41.2	45	57 ± 18.4	35.6

**Table 2 jcm-11-00769-t002:** Baseline echocardiographic characteristics of patients assessed in the included studies and differences between the groups.

Variables	CCM	IF	Healthy Controls	CCM vs. IF	IF vs. Controls
Mean age	50.8 (43.8, 57.8)	50.4 (45, 56)	44.3 (38.5, 50.2)	0.312 (−0.02, 0.64. I^2^ = 72%)	0.391 (0.06, 0.72. I^2^= 72%)
Males proportion (%)	49 (42, 56)	39 (31, 48)	44 (39, 50)	*p*-value = 0.213	*p*-value = 0.098
LVEF (%)	58.4 (51.5, 65.4)	63.3 (58.4, 68.2)	64.8 (60.6, 68.9)	−1.169 (−1.88, −0.46. I^2^ = 83%)	0.063 (−0.19, 0.32. I^2^ = 0%)
LAVI (ml/m^2^)	33.3 (27.6, 38.8)	-	-	-	-
LAD (mm)	35.8 (32.5, 39.2)	34.5 (30.6, 38.5)	34.6 (31.2, 37.9)	0.608 (0.05, 1.17. I^2^ = 43%)	0.115 (−0.29, 0.52. I^2^ = 9%)
LVD (mm)	53.3 (50.8, 55.9)	48.2 (43.9, 52.6)	49.2 (45.9, 52.4)	1.559 (0.45, 2.67. I^2^ = 92%)	0.022 (−0.28, 0.32. I ^2^= 21%)
LVS (mm)	34.7 (30.4, 38.9)	30.4 (25.9, 34.8)	30.3 (27.8, 32.8)	1.106 (0.37, 1.84. I^2^ = 84%)	−0.030 (−0.39, 0.33. I^2^ = 42%)
E/e’ ratio	8.2 (5.6, 10.9)	7.1 (3.6, 10.5)	6.6 (4.5, 8.7)	-	-
GLS (%)	−15.9 (−18.1, −13.7)	−19.9 (−21.2, −18.7)	−20.1 (−20.8, −19.3)	1.253 (0.53, 1.98. I^2^ = 94%)	0.197 (−0.19, 0.59. I^2^ = 80%)

## Data Availability

Not applicable.

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
