# Peer review of "Longitudinal Speckle Tracking Strain Abnormalities in Chagas Disease: A Systematic Review and Meta-Analysis"

_jcm, 2022, doi:10.3390/jcm11030769_

Round 1
Reviewer 1 Report
From 1044 references the authors selected ten studies comprising 1222 participants divided in 447 with Chagas cardiomyopathy (CCM), 444 of the "indeterminate" form (IF), and 301 healthy controls. Pts with CCM had higher mean global longitudinal strain than the IF, while no significant difference between the IF group with the healthy controls. But segmental strain analysis disclosed that that subjects with the IF form had significant worse strain values at the basal-inferoseptal and mid-inferoseptal segments in comparison with the healthy controls. As radial and circumferencial strain may also be abnormal it may be suggested to include as well these parameters. Further research with this novel technique should be pursued.
The findings by different authors in subjects with the "indeterminate form" of abnormal values of longitudinal strain in two segments compared with the healthy controls, despite comparable GLS, open the possibility of further research in this area.
Reviewer 2 Report
Most clinicians always assumed that indeterminate form in Chagas Disease (CD) is not a static phase, and the newer and more sensitive imaging techniques might identify disease progression. That said, this article helps understanding physiopathology of the disease, and enhances longitudinal strain as a potential assessment tool for re-stratification and treatment.
The article is of great interest and well written. However, modifications must been made:
MAJOR ISSUE
CCM patients have a higher rate of conduction abnormalities (https://www.ncbi.nlm.nih.gov/pmc/articles/PMC5999094/), and lots of them are paced.
In other diseases, patients with bundle branch blocks or ventricular pacing have abnormal values of GLS due to delayed depolarization and how deformation is sensed by the software in areas with delayed depolarization and contraction. This SR and MA as well as the papers that compared CCM vs IF don´t clearly explain how pacing or bundle branch blocks affected GLS. A clear explanation of how RBBB, LBBB, LAFB, and paced rhythm were considered, included or not, or how they affected GLS in each of the 10 articles should be included in results, tables (for example, adding a column with % of pts with conduction delays or paced), discussion, and limitations.
MINOR ISSUES
Intro
“CD currently affects approximately eight million people”
Please check Echeverria et al (cite 8) where current estimates are 6-7 million
“..remain in this stage without exhibiting organ involvement signs or symptoms”
I think you should better use “without exhibiting signs or symptoms of organ involvement”
“could, in addition, be critical for identifying patients with early myocardial involvement and those at risk of developing CCM”
Ventricular arrhythmias and sudden cardiac death are frequent and even present in patients with normal ejection fraction in CD. Could LS also be useful for indentifying patients at risk of developing arrhythmias? if so, clarify and cite.
Results
Indeterminate phase vs healthy controls
For the comparison between IF and healthy controls, a significative difference in the age of the groups was found, being healthy controls younger. This should be pointed out in the discussion, and promote investigators to use matched (Age-sex) healthy controls for further comparisons
I2 heterogeneity is high in most analyses. Please add an explanation and limitations in the discussion.
Discussion
First paragraph: first point out GLS diff in CCM vs IF, no signiff diff in IF vs healthy, as that was the main objective, and then point out the segments significantly affected.
…basal membrane thickening instead of “basement”
….and even LV segmental wall motion “abnomalities”
Last paragraph of discussion shouldn´t start with “However”
Availability, reliability and affordability of strain in rural areas of LMICs where most CD pts live is a limitation. We can´t redefine the disease course using a method unsuitable for the affected patients. This aspects could be addressed as a part of the discussion
